# Modelling and Predicting the Growth of *Escherichia coli* and *Staphylococcus aureus* in Co-Culture with *Geotrichum candidum* and Lactic Acid Bacteria in Milk

**Pavel Ačai** [1], **Martina Koňuchová** [2] and **Ľubomír Valík** [2,*]

1 Institute of Chemical and Environmental Engineering, Faculty of Chemical and Food Technology, The Slovak University of Technology Bratislava, 812 37 Bratislava, Slovakia; pavel.acai@stuba.sk
2 Institute of Food Sciences and Nutrition, Faculty of Chemical and Food Technology, The Slovak University of Technology Bratislava, 812 37 Bratislava, Slovakia; martina.konuchova@stuba.sk
* Correspondence: lubomir.valik@stuba.sk; Tel.: +421-918-674-518

**Abstract:** The growth of two pairs of co-cultures (*Escherichia coli*/*Geotrichum candidum* and *Staphylococcus aureus*/*Geotrichum candidum*) with a starter culture of lactic acid bacteria was studied in milk at temperatures ranging from 15 °C to 21 °C, related to the ripening of artisanal cheese. For an inoculum of approximately $10^6$ CFU/mL, LAB not only induced an early stationary phase of *E. coli* (two isolates BR and PS2) and *S. aureus* (isolates 2064 and 14733) but also affected their death phase. In co-cultures with LAB and *G. candidum*, the numbers of *E. coli* and *S. aureus* increased in 2 logs and 1 log, respectively, reaching maximum population densities (MPDs) of less than 5 and 4 logs, respectively. After that, the populations of both bacteria represented with two isolates decreased in more than 2 logs and 3 logs within 2 days compared to their MPDs, respectively. *G. candidum* was found to be the subject of interactions with LAB within a given temperature range only partially. To develop a tertiary model for the growth curves of the populations, a one-step approach was used, combining the Huang-Gimenez and Dalgaard primary model with secondary square-root models for growth rate and lag time. Furthermore, the reparametrized Gompertz-inspired function with the Bigelow secondary model was used to describe the death phase of the *E. coli* and *S. aureus* isolates. The prediction ability of the growth of the H-GD tertiary model for co-cultures was cross-validated within the isolates and datasets in milk and milk medium with 1% NaCl. The study can be used as knowledge support for the hygiene guidelines of short-ripened raw milk cheeses, as was our case in Slovakia.

**Keywords:** pathogens in co-cultures; *Escherichia coli*; *Staphylococcus aureus*; *Geotrichum candidum*; lactic acid bacteria

## 1. Introduction

A wide range of interactions among microbial populations belongs to actual and challenging subjects of quantitative food microbiology, especially when the research results have applicable predictive potential. Experiments referring to the fate of populations in the background of ongoing fermentation and ripening may provide a substantial view to several questions that usually appear in the practice of artisan cheese. For example, we may ask whether bacterial starters are more suitable and effective against contaminants than naturally present LAB populations with their diversity, actual activity, and acidification ability.

In predictive microbiology, two- and one-step approaches are usually applied to describe and predict growth, inactivation/survival, and interactions between microorganisms in foods with known environmental factors. First, the traditional two-step approach is based on the estimation of the pertinent kinetic parameters using a suitable primary model. Then, the effect of environmental conditions on these parameters is modelled using secondary models, and finally, their integration into a tertiary model enables the prediction

of microbial behaviour over time [1,2]. One-step data analysis is applied to construct the tertiary model directly with a combination of primary and secondary models, thus minimising global error [3]. For the co-culture description, various competitive/interaction models can be used. They are based on a semi-mechanistic approach, descriptive models such as Lotka-Voltera types, and models quantifying the "Jameson effect" [4,5]. Both two- and one-step approaches can be implemented to study the competitive growth of microbial populations. In this work, the one-step approach is preferred to model and predict the growth of *E. coli*, *G. candidum* and *S. aureus*, *G. candidum* in the presence of lactic acid bacteria in milk.

Except for *L. monocytogenes*, which is not the subject of this research, *E. coli* and *S. aureus* should be included as pathogenic bacteria considered a major safety issue for raw milk cheese. They are most often found in cheeses produced with raw milk and function as an indicator of hygiene deficiencies [6–8]. According to Desmarchelier and Fegan [9], raw milk can become contaminated with *E. coli* directly through animal faeces or indirectly through contaminated farm and dairy environments, equipment, and handling personnel. Although most *E. coli* strains are harmless commensals, some are known to be pathogenic bacteria, causing severe intestinal and extraintestinal diseases in humans. *The presence of S. aureus* in cheese is associated with post-secretory contamination and it is relevant as it may produce enterotoxins [10–12]. In general, the microbiota of artisanal cheese consists of complex assemblages consisting of not only prokaryotic but also fungal populations. As an example, the presence of *G. candidum* has been linked to the microbial profile of several artisanal raw milk cheeses from various countries [13–16].

Data on raw milk cheese quality and the prevalence of food-borne pathogens are well documented. However, there is less information in the literature on the in-depth knowledge of interactions between microbial populations [17,18]. On the one hand, physiological studies contribute to a better understanding of the behaviour of the microbiota, but studies in predictive microbiology go one step further by modelling and simulating microbial dynamics over time. Therefore, both can provide reproducible complex patterns that give insight into the effect of varying processing environmental conditions on the cheese microsystem [19–21]. As for most artisanal cheeses, ambient temperatures from 15 to 21 °C are applied during fermentation and the early phase of the ripening processes [10,22], our objective was to identify the microbial interactions between a starter culture of lactic acid bacteria, and isolates of *E. coli*, *S. aureus*, and *G. candidum* in detail, thus contributing to the knowledge of the raw milk cheesemaking. In addition, the other objectives are concerned with the predictive ability to evaluate and validate the proposed tertiary model for the behaviour of co-cultures in Slovakian lump cheese that is produced in mountain areas and sent for industrial processing to Bryndza cheese [23].

## 2. Material and Methods

### 2.1. Microorganisms and Culture Conditions

The commercial LAB culture DVS® Fresco® 1000NG and isolate J of *G. candidum* were used during all co-cultivation experiments. Mesophilic starter culture consisting of *Lactococcus lactis* subsp. *lactis*, *L. lactis* subsp. *cremoris*, and *Streptococcus salivarius* subsp. *thermophilus* was kept frozen at −45 °C. *G. candidum* was refrigerated at 5 °C on plate count skim milk agar slants (SMA; Merck, Darmstadt, Germany) and periodically sub-cultured in diluted SMA agar. There were four different series of *E. coli* and *S. aureus* co-cultivations using 2 isolates of each, BR, PS2 for *E. coli*; and 2064, 14733 for *S. aureus*. All bacterial cultures were maintained in Brain Hearth Infusion (BHI) broth (Sigma-Aldrich, St. Louis, MO, USA) at 5 ± 0.5 °C before analysis.

Fungal and bacterial cultures and isolates (Table 1) belong to the collection of the Institute of Food Science and Nutrition (the Slovak University of Technology in Bratislava, Bratislava, Slovakia). Their identifications were performed or confirmed in the previous works [24–26].

**Table 1.** Microbial cultures and their origin.

| Microorganism | Isolate | Origin |
|---|---|---|
| DVS® Fresco® 1000NG | - | commercial LAB culture; Christian Hansen, Hoersholm, Denmark |
| *G. candidum* | J | Slovakian traditional cheese "Bryndza" |
| *E. coli* | Br | Slovakian traditional cheese "Bryndza" |
| | PS2 | laboratory-produced pasta-filata cheese from raw cows' milk |
| *S. aureus* | 2064 | Slovakian ewes' lump cheese |
| | 14733 | milk vending machine biofilm |

### 2.2. Preparation of Inoculum and Experiments

Standard suspension of Fresco culture was prepared by inoculation of frozen culture into 100 mL of sterile milk and incubation at $30 \pm 0.5\ °C$ for 5 h until the stationary phase was reached. Standard suspension of *G. candidum* isolate was prepared from 48 h old culture grown on vertical SMA agar at $30 \pm 0.5\ °C$ and mixed with 10 mL of sterile saline solution. A standard suspension of *E. coli* and *S. aureus* isolates was prepared from a 24 h old culture grown in BHI broth at 37 °C. The above inoculation procedure was aimed to reach the initial concentration of Fresco at $10^6$ CFU/mL, of *G. candidum* at approximately $10^2$ CFU/mL, and *E. coli* and *S. aureus* isolates at approximately $10^3$ CFU/mL.

All co-cultivation experiments were performed in 250 mL of pre-tempered ultra-high temperature treated milk with 1.5 g/L fat content (Rajo, Ltd., Bratislava, Slovak Republic) without or with 1% NaCl ($w/v$). The incubation was performed in three parallel stages under static conditions at temperatures of 15, 18 and $21 \pm 0.5\ °C$, which represent artisanal cheese production [22].

The pH was measured using a WTW Inolab 720 pH-meter (Inolab, Weilheim, Germany) equipped with a SenTix 81 glass electrode (WTW GmbH, Weilheim, Germany) with the same time interval as samples for microbiological quantification. $a_w$-values were estimated by the LabMaster-aw (Novasina, Lachen, Switzerland).

### 2.3. Quantification of Microorganisms

The counts of LAB, *G. candidum*, *S. aureus* and *E. coli* were determined by the 10-fold dilution method in a saline-peptone solution. To achieve the best possible fit of the model to the curve, time intervals were predefined according to the incubation temperature.

Counts of cocci from Fresco culture were determined on M17 agar (Merck, Darmstadt, Germany) after 48 h incubation at $30 \pm 0.5\ °C$ according to EN ISO 15214 [27]. *G. candidum* counts were determined on DRBC agar (Biokar Diagnostics, Beauvais, France) after 5 days of incubation at $25 \pm 0.5\ °C$ according to EN ISO 21527-1 [28]. *E. coli* was counted on Chromocult Coliform agar (Merck, Darmstadt, Germany) after 24 h incubation at $37 \pm 0.5\ °C$ according to the National Standard Method F23 [29]. *S. aureus* was enumerated on Baird-Parker agar (Merck, Darmstadt, Germany) with incubation at $37 \pm 0.5\ °C$ for 48 h according to EN ISO 6888-1 [30].

### 2.4. Mathematical Models

2.4.1. Modelling the Microbial Interaction in Co-Cultures

The primary models of Huang [31] and Giménez and Dalgaard [5] combined with secondary square root models as applied for growth rate and lag time were used to describe competitive growth of the co-cultures series in milk for all isothermal growth curves. The suggested interaction H-GD model with competition coefficients describing the growth of LAB, *G. candidum* and behaviour of *E. coli* and *S. aureus* in inter-species competition were used in this study according to Ačai et al. [19]. Thus, the system of the ordinary differential equations with the initial conditions applied for the growth phase and a mixed system of

differential equations and nonlinear algebraic equation (G-B model) was used for survivors of *S. aureus* (*Sa*) and *E. coli* (*Ec*) in the death phase, denoted here as index *P* (*P = Ec* or *Sa*). The equations can be written as follows:

A. H-GD model with the competition coefficients

Growth of *LAB*; $t \in \langle 0, t_t \rangle$

$$\frac{dx_{Lab}}{dt} = \left[ \frac{\frac{\mu_{\max}^{Lab}}{\ln 10}}{1 + 10^{-\alpha(t-t_\lambda^{Lab})}} \left( 1 - 10^{(x_{Lab} - x_{\max,Lab})} \right) \left( 1 - 10^{(x_P - x_{\max,P})} \right) \right] I_{LP} \tag{1}$$

Growth of *P = Ec* or *Sa*; $t \in \langle 0, t_t \rangle$

$$\frac{dx_P}{dt} = \left[ \frac{\frac{\mu_{\max}^{P}}{\ln 10}}{1 + 10^{-\alpha(t-t_\lambda^{P})}} \left( 1 - 10^{(x_P - x_{\max,P})} \right) \left( 1 - 10^{(x_{Lab} - x_{\max,Lab})} \right) \right] I_{PL} \tag{2}$$

Survival *P = Ec* or *Sa*; $t \in \langle t_t, t \rangle$

$$x = x_{\max,P} + (x_{res,P} - x_{\max,P}) \exp \left\{ -\exp \left[ \left( \frac{-\frac{k_{\max,P}}{\ln 10} \cdot e}{(x_{res,P} - x_{\max,P})} \right) (t_{\lambda,s} - t) + 1 \right] \right\} \tag{3}$$

$$\frac{dx_{Gc}}{dt} = \frac{k_{Gc} \frac{\mu_{\max}^{Gc}}{\ln 10}}{1 + 10^{-\alpha(t-t_\lambda^{Gc})}} \left( 1 - 10^{(x_{Gc} - x_{\max,Gc})} \right) \tag{4}$$

$$t = 0 \quad x_{Lab} = x_{Lab,0} \quad x_P = x_{P,0} \quad x_{Gc} = x_{Gc,0}$$

where $\mu_{\max}^{i=Lab,P,Gc}$ are the maximum specific growth rates of the LAB, *E. coli* or *S. aureus* and *G. candidum* (*Gc*), respectively, $t_\lambda^i$ are the lag times of microorganisms, $\alpha$ is the lag phase transition coefficient, taking a value of 4 [31]. Concentrations $x$, which include $x_i = \log N_i$, $x_{0,i} = \log N_{0,i}$, $x_{\max,i} = \log N_{\max,i}$, $x_{res,i} = \log N_{res,i}$ represent the real, initial, maximum and residual (or tail) cell density, $N_i$, $N_{0,i}$ $A_{\max,i}$ and $N_{res,i}$. $I_{LP}$, $I_{PL}$ are the competition coefficients representing the effects of LAB (Fresco) on *E. coli* or *S. aureus* and *E. coli* or *S. aureus* on LAB (Fresco), respectively, in the H-GD model type R. $k_{\max,P}$ is the maximum death rate of *E. coli* or *S. aureus*, $t_{\lambda s}^{i=Ec, Sa}$ is the survival curve shoulder, $t_t$ is the transitioning breakpoint time from stationary to survival phase for *E. coli*/*S. aureus* that is determined so that the time $t_\lambda$ is equal to zero, $k_{Gc}$ is the reduction coefficient for the *G. candidum* growth rate.

As $\mu_{\max}^i$ and $t_\lambda^i$ are a function of temperature, the following secondary square root models were used to incorporate the effect of temperature on growth parameters [32]:

$$\sqrt{\mu_{\max}^i} = b_{T,i} \cdot (T - T_{\min,i}) \tag{5}$$

where regression coefficient $b_T$ (h$^{-1}\cdot{}^\circ$C$^{-1}$) is the slope and depends on additional growth conditions and the microorganism involved, $T$ ($^\circ$C) is the temperature, and $T_{\min}$ is its theoretical minimum for growth.

$$\mu_{\max}^{Ec} = [b_{T,Ec} \cdot (T - T_{\min,Ec})]^2; \ \mu_{\max}^{Sa} = [b_{T,Sa} \cdot (T - T_{\min,Sa})]^2; \ \mu_{\max}^{Lab} = [b_{T,Lab} \cdot (T - T_{\min,Lab})]^2;$$
$$\mu_{\max}^{Gc} = [b_{T,Gc} \cdot (T - T_{\min,Gc})]^2 \tag{6}$$

The square root relation between lag time ($t_\lambda$) and *T* was used in the H-GD models according to [33]:

$$t_\lambda^{Ec} = \frac{1}{[b_{\lambda,Ec} \cdot (T - T_{\min,Ec})]^2}; \ t_\lambda^{Sa} = \frac{1}{[b_{\lambda,sa} \cdot (T - T_{\min,Sa})]^2}; \ t_\lambda^{Lab} = \frac{1}{[b_{\lambda,Lab} \cdot (T - T_{\min,Lab})]^2};$$
$$t_\lambda^{Gc} = \frac{1}{[b_{\lambda,Gc} \cdot (T - T_{\min,Gc})]^2} \tag{7}$$

where $b_{\lambda,I}$ is the regression coefficient.

Next, for the declination phase of *E. coli* and *S. aureus* isolates, the reparametrized Gompertz-inspired survival model together with the Bigelow secondary model was used for its versatility of fitting linear data and those that have shoulder and/or tailing effects [34]. The $z_i$ parameter was used to help in the theoretical description of the influence of temperature and other factors acting in this phase such as the pH drop and addition of NaCl.

The secondary Bigelow log-linear model was applied to express the dependence of the rate of decrease, $k_{max}$ on temperature as follows:

$$k_{max}^{Ec} = \frac{k_{ref}^{Ec}}{10^{\frac{T_{ref}-T}{z_{Ec}}}}; \; k_{max}^{Sa} = \frac{k_{ref}^{Sa}}{10^{\frac{T_{ref}-T}{z_{Sa}}}} \tag{8}$$

where $r_{efi}^{t}$ is $k_{max}^{i}$ at a reference temperature ($T_{ref}$) and $z$ represents a temperature required for a 10-fold reduction of *E. coli* or *S. aureus* numbers.

2.4.2. Parameter Determination and Evaluation of Model Performance

The one-step kinetic data analysis method described by Huang [35] was applied for parameter optimisation from the given isothermal growth curves of the co-culture microbial populations. To construct the tertiary H-GD model and minimise the global sum of squared errors (*SSE*), they were analysed concurrently using the H-GD models (Equations (1)–(4)) and secondary square root models (Equations (5)–(7)).

In the beginning, the tertiary H-GD model had 19 parameters (Equations (1)–(8)): three average maximum values of $x_{max}$, competition coefficients $I_{LP}$, $I_{PL}$; and reduction coefficient $k_{Gc}$ in the H-GD model type R and reduction coefficient $k_{Gc}$ (Equations (1)–(4)) and parameters $b_T$, $T_{min}$, $b_\lambda$ from the square-root models (Equations (5)–(7)). The parameters $N_{res}$, $k_{ref}$, and $z$ are derived from the reparametrized Gompertz-inspired survival model (Equation (3)), as well as the Bigelow secondary model (8), respectively.

$$p_{H-GD} = \left\{ \begin{array}{l} x_{max,Lab}; x_{max,P}; x_{max,Gc}; I_{LP}; I_{PL}; k_{Gc} \\ b_{T,Lab}; b_{T,P}; b_{T,Gc}; T_{min,Lab}; T_{min,P}; T_{min,Gc}; b_{\lambda,Lab}; b_{\lambda,P}; b_{\lambda,Gc} \\ k_{ref,P}; N_{res,P}; z_P \end{array} \right\} \tag{9}$$

$p_{H-GD}$ is the vector of parameters of H-GD models for the simultaneous competitive growth of the co-culture series.

The prediction ability of the tertiary H-GD model was tested for a reduced number of parameters ($p_E$): the average maximum density counts of microorganisms ($x_{max,Lab}$, $x_{max,P}$, $x_{max,G}$), competition coefficients ($I_{LP}$, $I_{PL}$), reduction coefficient $k_{Gc}$, regression coefficient, $b_{\lambda,Gc}$, maximum declination rates at a reference temperature $T_{ref}$ ($k_{ref,Ec}$, $k_{ref,Sa}$), and $z$-values ($z_{Ec}$, $z_{Sa}$). Regression coefficients $b_{\lambda,Ec}$ for the isolate *E. coli* PS2 were also evaluated by using one-step kinetic data analysis. The remaining parameters in Equation (8), which were previously optimised by nonlinear regression analysis for single cultures [19] or taken from the following scientific articles [24,36], were fixed as constants. This approach has the advantage that some parameters for co-culture growth prediction in milk could be estimated from the growth of individual species.

The goodness of fit of the tertiary H-GD model was evaluated with the global sum of squared errors (*SSE*), the root mean square error (*RMSE*), and the determination coefficient ($R^2$) to evaluate its suitability to fit the whole set of observation points according to Equations (1)–(8).

$$SSE = \sum_{i=1}^{n} \left( x_i^{exp} - x_i^{cal} \right)^2 \tag{10}$$

$$RMSE = \sqrt{\frac{\sum\limits_{i=1}^{n}\left(x_i^{\exp} - x_i^{cal}\right)^2}{n-p}} \qquad (11)$$

$$R^2 = 1 - \frac{SSE}{SST} \qquad (12)$$

where $x_i^{\exp}$ and $x_i^{cal}$ correspond to the observed and predicted values, respectively, $n$ is the total number of data points, $p$ is the number of estimated parameters, and $SST$ is the total sum of squared errors.

The prediction capability of the H-GD tertiary model was tested through the bias ($B_f$) and accuracy ($A_f$) factors [37] on the datasets of different *E. coli* (BR and PS2) and *S. aureus* (2064 and 14733) isolates within the temperature range of 15 to 21 °C for the cases without and with 1% NaCl addition

$$B_f = 10^{\frac{\sum\limits_{i=1}^{n}\left(\log x_i^{cal} - \log x_i^{\exp}\right)}{n}} \qquad (13)$$

$$A_f = 10^{\sqrt{\frac{\sum\limits_{i=1}^{n}\left(\log x_i^{cal} - \log x_i^{\exp}\right)^2}{n}}} \qquad (14)$$

The accuracy of the H-GD model was checked with the *RMSE* of prediction for each microorganism in the co-culture growth according to [38].

The tertiary model parameters were estimated using the commercial process-engineering software Athena Visual Workbench (Stewart & Associates Engineering Software, Madison, WI; www.athenavisual.com (accessed on 15 November 2022)). *SSE*, *RMSE*, as well as bias ($B_f$) and accuracy ($A_f$), were calculated using Microsoft Excel (Microsoft, Redmond, WA, USA).

## 3. Results and Discussion

Cheese ecosystems may be associated with the presence of unique microbes, leading to microbial interactions that can develop remarkable sensory characteristics. The idea of providing quantitative studies or proofs related to the role of microbial interactions in the microbiological quality and safety of artisanal raw milk products was inspired not only by the history of their production but also by the recent scientific outputs published [10,39–42]. According to Schoustra et al. [43], traditionally processed foods derived from raw milk may be safe, since adjustment of the doses of the LAB starters can serve as a means of controlling sanitary protection, maintaining bacterial diversity [11,44], and supporting the activity of inherited populations of LAB as well [45]. This is also the case of artisanal lump cheese produced in Slovakian mountain areas and sent for processing for Bryndza cheese after a short 8 to 10 days of ripening [23].

### 3.1. One-Step Analysis of Competitive Growth

By combining co-culture primary growth models with secondary growth and survival models in one step, we were able to describe the growth patterns of three microbial populations in this work. As presented in Figure 1 for *E. coli* BR in milk and milk with 1% NaCl, respectively, the following similar characteristics can be recognised. First, LAB played a dominant role in co-cultures; grown at the highest rates that were influenced mostly by the temperature and only partly by 1% NaCl content. Dalcanton et al. [46] and Medveďová et al. [47] reported a comparable trend regarding the influence of temperature and water activity on the behaviour of LAB. On average, they reached maximum population density (MPD) of $9.32 \pm 0.07$ logs and a stable population increase (the difference between MPD and $N_0$) of $3.2 \pm 0.3$ log CFU/mL in all co-culture trials in the shortest time. These results aligned with our previous work [19] and those reported for the co-culture growth of *Lactiplantibacillus plantarum* with *S. aureus* by Rodríguez-Sánchez et al. [48].

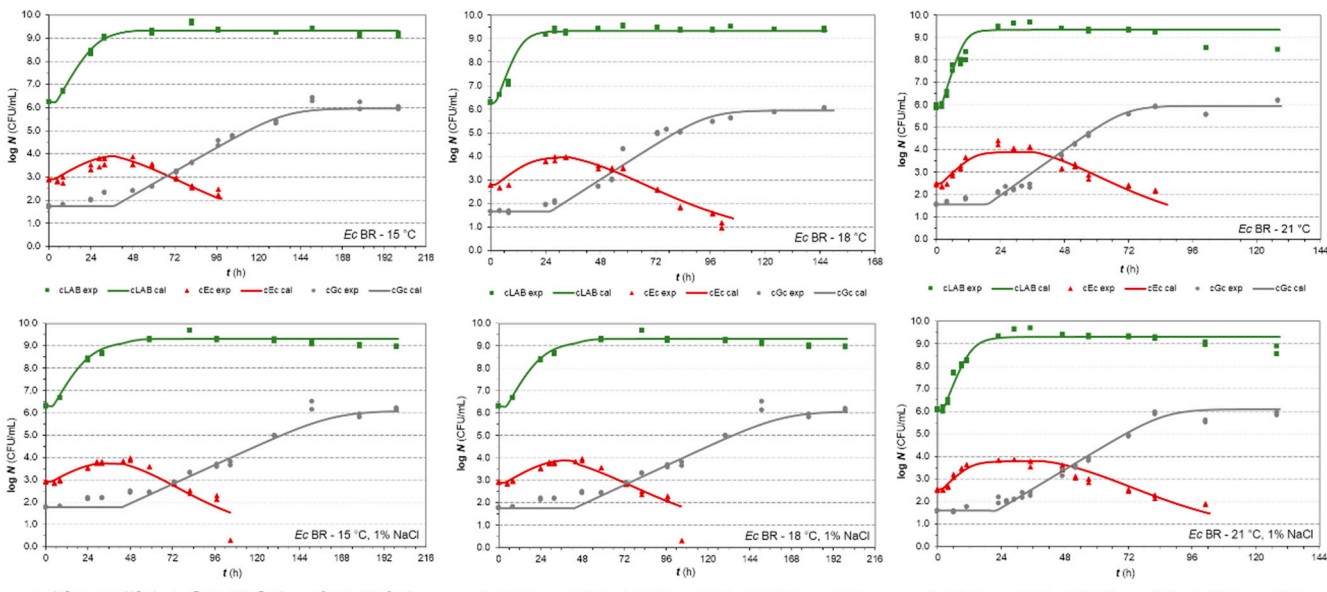

**Figure 1.** Co-culture growth of LAB Fresco, *E. coli* BR, and *G. candidum* in milk at 15, 18, and 21 °C (without and with 1% NaCl, respectively). The continuous lines represent the growth predicted values by the H-GD model and the dots represent the experimental values (■ LAB, ▲ Ec, ● Gc).

While grown independently, LAB determined the responses of *E. coli* BR that increased the numbers in 1.23 ± 0.32 log CFU/mL for a period in which LAB reached the early stationary phase. *E. coli* reached its stationary phase with a maximum population density (MPD) of 3.91 ± 0.18 logs on average and prolonged its duration with increasing temperature in both cases of milk (without and with 1% NaCl). Consistent with these results, Sreekumar and Hosono [49] reported final counts of *E. coli* 3544 less than 3 logs CFU/mL during co-cultivation with *Lactobacillus acidophilus* SBT2071 in semi-skimmed sheep's milk. Referring to the increase in population, this was not the case for *G. candidum.* However, in milk, as well as at 1% NaCl, its population reached the highest increase of 4.19 ± 0.16 logs, for a longer period than other populations. Naturally, this period was determined by temperature. It can also be seen that the *G. candidum* lag phase was almost identical to the period that covers the LAB lag and exponential phases together.

The growth studies [24,50] demonstrated a similar pattern of growth responses during *G. candidum* and LAB Fresco co-culturing experiments. During the stationary phase, the yeast was able to grow exponentially and reach its stationary phase. This can be explained by both lactate consumption and ammonium production [51], which are related to an increase in pH and tolerance to lactic acid produced by LAB.

Similar responses were also found in co-culture trials, in which the previous isolate of *E. coli* was replaced by the PS2 isolate isolated from Slovakian artisanal steamed and stretched Slovakian cheese. Compared to BR isolate, the only visually recognisable differences are referred to as a higher MPD and population increase of 4.76 ± 0.26 log and 1.91 ± 0.33 log, respectively, as well as longer durations of stationary phases in general (Supplementary Figure S1). Thus, naturally, the PS2 isolate also grew at a negligible higher rate than the BR isolate during the exponential phase.

The *S. aureus* isolate 14733 in the same type of co-cultures with LAB and *G. candidum* reached MPD and a population increase of 3.97 ± 0.36 and 1.06 ± 0.29 log, respectively (Figure 2). Both mean values were close to the responses of the *E. coli* BR isolate.

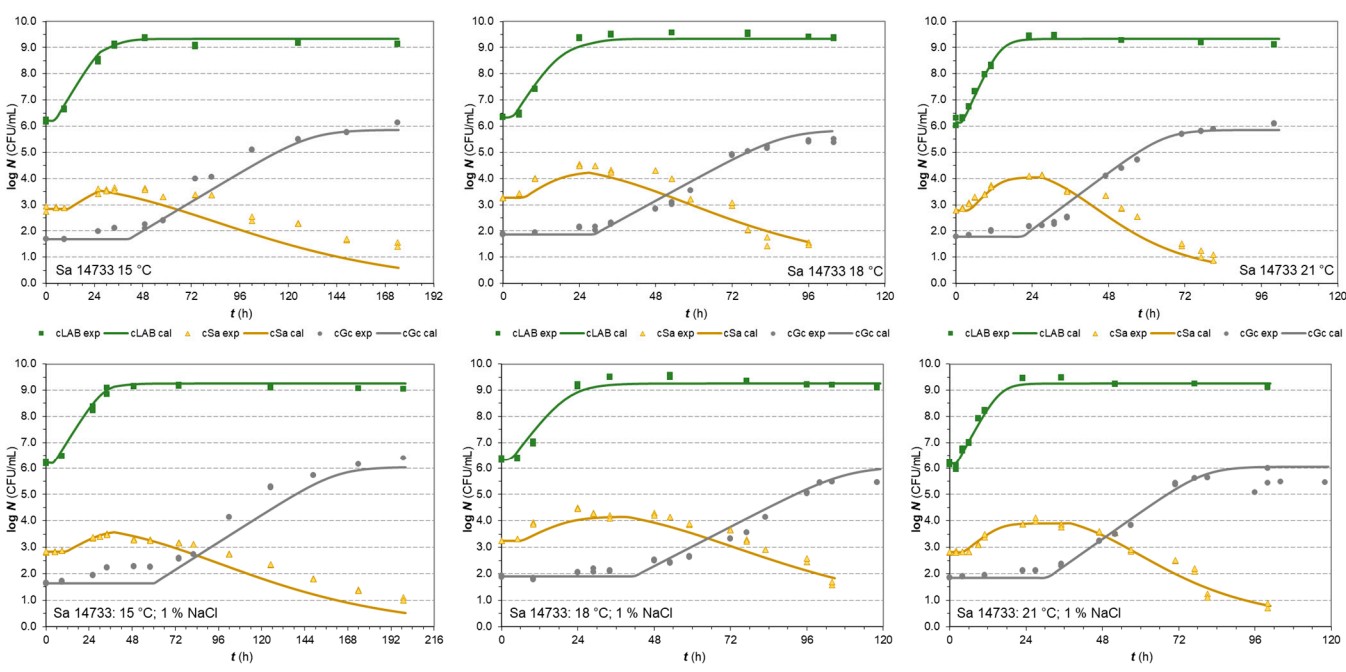

**Figure 2.** Co-culture growth of LAB Fresco, *S. aureus* 14733, and *G. candidum* in milk at 15, 18, and 21 °C (without and with 1% NaCl, respectively). The continuous lines represent the growth predicted values by the H-GD model and the dots represent the experimental values (■ LAB, ▲ Sa, ● Gc).

The second *S. aureus* isolate 2064 [52] was also used in the last series of co-cultures with LAB and *G. candidum*. The results of the tests (Supplementary Figure S2) indicated that this isolate appeared to be sensitive to lactic acid or competition in general since it reached MPD in the stationary phase in less than 4 logs (3.38 ± 0.40 log CFU/mL) and the increase in population in less than 1 log (0.65 ± 0.27 log CFU/mL).

The production of organic acids has a major impact on the quality of the final products during the cheesemaking process. Acidification is usually achieved by the production of lactic acid through the fermentation of lactose by LAB [48,53]. The pH changes in our study followed a sigmoidal behaviour throughout LAB growth but with a delay of approximately 6 to 10 h. This trend is consistent with those reported by Rodríguez-Sánchez et al. [48] when analysing the antimicrobial activity of the LABs against some potentially pathogenic bacteria used as indicators, including *S. aureus*. As LAB growth progressed to the exponential and stationary growth phase, a rapid and significant drop in pH (pH ≤ 5.3) was observed in the second phase.

The parameters estimated for the H-GD model are presented in Tables 2 and 3 for both *E. coli* and *S. aureus* isolates in co-cultures, respectively.

Except for the facts described above, the data in Table 2 pointed out that the competition coefficients $I_{EL}$ (Equation (2)) in the H-GD model showed a similar growth reduction (<60%) for both isolates of *E. coli* compared with their original capacity as individual species in milk [19]. On the other hand, the competitive effect of LAB on *S. aureus* isolates was stronger and strain-dependent. The coefficients $I_{SL}$ in Table 3 were significantly lower and different for isolates 2064 and 14733. Thus, they confirmed the higher sensitivity to non-specific inhibition caused by LAB that was found for isolate 14733.

**Table 2.** Parameters of the H-GD model with 95% highest posterior density interval (Bayesian estimation) for growth of *E. coli* (isolates BR and PS2) in co-cultures with *G. candidum* and LAB in milk.

| Parameters | *E. coli* (Isolate BR) | | *E. coli* (Isolate PS2) | |
|---|---|---|---|---|
| | In Milk | 1% NaCl in Milk | In Milk | 1% NaCl in Milk |
| $x_{\max,\text{Lab}}$ | $9.34 \pm 0.04$ | $9.32 \pm 0.03$ | $9.36 \pm 0.04$ | $9.33 \pm 0.04$ |
| $x_{\max,\text{Ec}}$ | $4.17 \pm 0.16$ | $3.95 \pm 0.10$ | $5.14 \pm 0.17$ | $5.14 \pm 0.10$ |
| $x_{\max,\text{Gc}}$ | $5.96 \pm 0.08$ | $6.09 \pm 0.10$ | $5.72 \pm 0.08$ | $6.04 \pm 0.17$ |
| $I_{\text{LE}}$ | $1.158 \pm 0.093$ | $1.254 \pm 0.100$ | $0.957 \pm 0.059$ | $0.951 \pm 0.054$ |
| $I_{\text{EL}}$ | $0.526 \pm 0.045$ | $0.536 \pm 0.049$ | $0.588 \pm 0.042$ | $0.513 \pm 0.035$ |
| $k_{\text{Gc}}$ | $0.850 \pm 0.038$ | $0.710 \pm 0.025$ | $0.931 \pm 0.048$ | $0.749 \pm 0.046$ |
| $k_{\text{ref}}$ | $0.101 \pm 0.006$ | $0.101 \pm 0.006$ | $0.133 \pm 0.006$ | $0.081 \pm 0.006$ |
| $x_{\text{res,Ec}}$ | $0.4$ [a] | $0.42 \pm 0.16$ | $1.20 \pm 0.29$ | $0.5$ [d] |
| $z_{\text{Ec}}$ | $30.67 \pm 5.68$ | $32.25$ [d] | $6.38 \pm 0.70$ | $28.21 \pm 5.76$ |
| $b_{\lambda,\text{Gc}}$ | $0.0109 \pm 0.0003$ | $0.0101 \pm 0.0003$ | $0.0096 \pm 0.0002$ | $0.0085 \pm 0.0002$ |
| $b_{\text{T,Gc}}$ [b] | $0.0228$ [b] | $0.0228$ [b] | $0.0228$ [a] | $0.0228$ [a] |
| $T_{\min,\text{Gc}}$ [b] | $0.00$ [b] | $0.00$ [b] | $0.00$ [a] | $0.00$ [a] |
| $b_{\lambda,\text{Lab}}$ [c] | $0.0343$ [c] | $0.0343$ [c] | $0.0343$ [b] | $0.0343$ [b] |
| $b_{\text{T,Lab}}$ [c] | $0.0384$ [c] | $0.0384$ [c] | $0.0384$ [b] | $0.0384$ [b] |
| $T_{\min,\text{Lab}}$ | $1.11$ [c] | $1.11$ [c] | $1.11$ [b] | $1.11$ [b] |
| $b_{\lambda,\text{Ec}}$ | $0.0493$ [c] | $0.0493$ [c] | $0.0365 \pm 0.0045$ | $0.0366 \pm 0.0044$ |
| $b_{\text{T,Ec}}$ | $0.0421$ [c] | $0.0421$ [c] | $0.052$ [c] | $0.052$ [c] |
| $T_{\min,\text{Ec}}$ | $4.16$ [c] | $4.16$ [c] | $4.80$ [c] | $4.80$ [c] |

Transition time from stationary to survival phase: $t_t$ (milk) = 36 h; $t_t$ (milk + 1% NaCl) = 44 h. [a] the parameter lower bound. [b] the parameter values are fixed [54]. [c] the parameter values are fixed [19]. [d] the parameter value is not determined.

The results in Tables 2–4 showed that the tertiary H-GD model successfully described the competitive growth between species in milk at temperatures related to the ripening conditions of artisanal cheesemaking. The global *RMSE* values for all cases are lower than 0.33 log CFU/mL, which is well within the range of normal experimental error. A dominant level of inoculum (approximately $10^6$ CFU/mL) of a starter culture favoured the growth of LAB in milk (the competition coefficient $I_{\text{LP}}$ is greater than one in almost all cases and was able not only to induce an early stationary state in *E. coli* (isolates BR and PS2) and *S. aureus* (isolates 2064 and 14733) for cases without and with 1% NaCl addition but also subsequently reduced their population. LAB growth of the LAB slightly suppressed the growth rate of *G. candidum* of its original ability as a single species in milk. The reduction coefficients of the growth rate of *G. candidum* $k_{\text{Gc}}$ were within the region 0.710–0.995. Naturally, their values were lower for the cases with NaCl addition (Tables 2 and 3).

To conclude the experimental findings, except for *L. monocytogenes*, the four decisive populations of cheese fermentation and ripening can be ranked according to their ability to compete with each other. Taking into account two isolates of *S. aureus* and *E. coli*, *S. aureus* showed less competitive behaviour, followed by *E. coli*. However, *G. candidum*, with its ability to assimilate lactic acid produced by LAB appears to be an independent player in co-cultures. Finally, it was confirmed that after proper milk acidification, LAB is (and should be) a dominant and stable population that, depending on the time, may control other undesirable microbial populations.

**Table 3.** Parameters of H-GD model with 95% highest posterior density interval (Bayesian estimation) for growth of *S. aureus* (isolates 2064 and 14733) in co-cultures with *G. candidum* and LAB in milk.

| Parameters | *S. aureus* (Isolate 2064) | | *S. aureus* (Isolate 14733) | |
|---|---|---|---|---|
| | **In Milk** | **1% NaCl in Milk** | **In Milk** | **1% NaCl in Milk** |
| $x_{\text{max,Lab}}$ | $9.43 \pm 0.03$ | $9.40 \pm 0.05$ | $9.34 \pm 0.03$ | $9.25 \pm 0.03$ |
| $x_{\text{max,Sa}}$ | $3.83 \pm 0.15$ | $4.17 \pm 0.11$ | $4.43 \pm 0.12$ | $4.43 \pm 0.16$ |
| $x_{\text{max,Gc}}$ | $5.65 \pm 0.12$ | $5.82 \pm 0.17$ | $5.85 \pm 0.11$ | $6.04 \pm 0.15$ |
| $I_{\text{LS}}$ | $1.262 \pm 0.056$ | $1.083 \pm 0.057$ | $1.064 \pm 0.044$ | $0.912 \pm 0.043$ |
| $I_{\text{SL}}$ | $0.308 \pm 0.144$ | $0.174 \pm 0.089$ | $0.705 \pm 0.079$ | $0.526 \pm 0.054$ |
| $c_{\text{LS}}$ | - | - | - | - |
| $c_{\text{SL}}$ | - | - | - | - |
| $k_{\text{Gc}}$ | $0.995 \pm 0.067$ | $0.778 \pm 0.058$ | $0.906 \pm 0.048$ | $0.850 \pm 0.055$ |
| $k_{\text{ref}}$ | $0.133 \pm 0.022$ | $0.102 \pm 0.007$ | $0.107 \pm 0.007$ | $0.094 \pm 0.006$ |
| $x_{\text{res,Sa}}$ | $1.47 \pm 0.13$ | $0.3$ [c] | $0.5$ [c] | $0.5$ [c] |
| $z_{\text{Sa}}$ | $9.46 \pm 1.21$ | $10.44 \pm 0.51$ | $11.49 \pm 1.18$ | $13.79 \pm 1.67$ |
| $b_{\lambda,\text{Gc}}$ | $0.0092 \pm 0.0002$ | $0.0086 \pm 0.0003$ | $0.0104 \pm 0.0003$ | $0.0086 \pm 0.0002$ |
| $b_{\text{T,Gc}}$ | $0.0228$ [a] | $0.0228$ [a] | $0.0228$ [a] | $0.0228$ [a] |
| $T_{\text{min,Gc}}$ | $0.00$ [a] | $0.00$ [a] | $0.00$ [a] | $0.00$ [a] |
| $b_{\lambda,\text{Lab}}$ | $0.0343$ [b] | $0.0343$ [b] | $0.0384$ [b] | $0.0384$ [b] |
| $b_{\text{T,Lab}}$ | $0.0384$ [b] | $0.0384$ [b] | $1.11$ [b] | $1.11$ [b] |
| $T_{\text{min,Lab}}$ | $1.11$ [b] | $1.11$ [b] | $0.0302$ [b] | $0.0302$ [b] |
| $b_{\lambda,\text{Sa}}$ | $0.0302$ [b] | $0.0302$ [b] | $0.0409$ [b] | $0.0409$ [b] |
| $b_{\text{T,Sa}}$ | $0.0409$ [b] | $0.0409$ [b] | $5.02$ [b] | $5.02$ [b] |
| $T_{\text{min,Sa}}$ | $5.02$ [b] | $5.02$ [b] | | |

Transition time from stationary to survival phase: $t_t$ (milk) = 23 h; $t_t$ (milk + 1% NaCl) = 23 h. [a] the parameter values are fixed [54]. [b] the parameter values are fixed [19]. [c] the parameter lower bound.

**Table 4.** Goodness-of-fit indices and models comparison of H-GD model for the *E. coli* and *S. aureus* isolates in co-culture with *G. candidum* and LAB Fresco in milk.

| Indices | *E. coli* BR | | *E. coli* PS2 | | *S. aureus* 2064 | | *S. aureus* 14733 | |
|---|---|---|---|---|---|---|---|---|
| | **in Milk** | **1% NaCl in Milk** | **in Milk** | **1% NaCl in Milk** | **in Milk** | **1% NaCl in Milk** | **in Milk** | **1% NaCl in Milk** |
| *SSE* | 14.719 | 16.080 | 19.450 | 25.719 | 10.625 | 11.725 | 15.184 | 17.592 |
| $R^2$ | 0.992 | 0.991 | 0.987 | 0.986 | 0.991 | 0.991 | 0.989 | 0.988 |
| *p* | 10 | 10 | 11 | 11 | 10 | 10 | 10 | 10 |
| *RMSE* | 0.251 | 0.254 | 0.289 | 0.324 | 0.280 | 0.284 | 0.270 | 0.282 |

Number of data points: $n_{\text{EC BR}}$ (milk) = 244; $n_{\text{EC BR}}$ (milk + 1% NaCl) = 254; $n_{\text{EC PS2}}$ (milk) = 244; $n_{\text{EC PS2}}$ (milk + 1% NaCl) = 256; $n_{\text{Sa 2064}}$ (milk) = 146; $n_{\text{Sa 2064}}$ (milk + 1% NaCl) = 155; $n_{\text{Sa 14733}}$ (milk) = 218; $n_{\text{Sa 14733}}$ (milk + 1% NaCl) = 232.

### 3.2. Model Validation

In fermentations, the most important populations consist of the LABs responsible for fermentation and some adjunct culture. In our case, they are represented by Fresco culture and *G. candidum*, respectively. Inviting intensive growth, they create or supply added value to the final characteristics of the fermented product. On the other hand, co-culture studies are usually aimed at the behaviour of microbial contaminants, while the fermentation and adjunct cultures are monitored in the background. Undesirable, pathogenic, or spoilage bacteria play different roles and, in this work, are represented by the isolates of *E. coli* or *S.*

*aureus*. To evaluate strain variability or validate the co-culture models, most of their points of view are considered in this section.

### 3.2.1. *E. coli* Isolates in Co-Cultures

First, within the variability evaluation, the *RMSE* values [38] between milk growth and milk with 1% NaCl were evaluated for each *E. coli* isolate. Although the PS2 isolate showed the highest difference in milk numbers between 1% and 0% (*RMSE* = 0.67 log CFU/mL), the *RMSE* for the BR isolate was only 0.37 log CFU/mL, indicating that this isolate was less sensitive to NaCl addition. In the evaluation of the differences between isolates BR and PS2 in the same medium, they were lower in milk without salt addition (*RMSE* = 0.75 log CFU/mL) than in milk with 1% NaCl (*RMSE* = 1.11 log CFU/mL). Therefore, these values are more about the different behaviour between *E. coli* isolates, competitiveness, and sensitivity to NaCl than about variability. The lowest calculated *RMSE* values for LAB in all combinations ranged between 0.22 and 0.30 log CFU/mL. For *G. candidum* sensitive to NaCl, higher *RMSE* values of 0.49–0.69 log CFU/mL were found between co-culture growth in milk without NaCl and milk with the addition of 1% NaCl.

The prediction capability of the H-GD model was also evaluated through the bias ($B_f$) and accuracy ($A_f$) factors [37] for each microorganism in the *E. coli* co-cultures. For *E. coli*, the HG-D model data of isolate BR were validated with the experimental data of isolate PS2. The calculated $B_f$ values for the growth of *E. coli* isolates were within 0.993–1.387 and the $A_f$ values ranged from 1.283 to 1.704. With high probability, the prediction of *E. coli* growth was affected by the growth ability and sensitivity of the PS2 isolate to the addition of NaCl.

LAB growth was accurately predicted for all co-culture cases and the calculated $B_f$ values were between 0.996 to 1.003, while the $A_f$ values ranged from 1.026 to 1.035 showing that, on average, the predicted value was 2.6 to 3.5% different (either smaller or greater) from the observed values. Accurate prediction of LAB growth also confirmed the fact that their growth was minimally affected by other co-culture populations such as *G. candidum* or *E. coli* as the values of the interaction coefficient $I_{LE} \cong 1.0$ indicated before (Table 2).

### 3.2.2. *S. aureus* Isolates in Co-Cultures

According to data in Table 3, the fast-growing isolate 2064 of *S. aureus* compared with the numbers of isolate 14733 showed a large difference in milk and milk with 1% NaCl (*RMSE* = 0.66 and 0.91 log CFU/mL, respectively). On the other hand, the differences within the same isolate but between different media (milk without NaCl vs. 1% NaCl) were lower in the case of halotolerant *S. aureus*, 0.39 and 0.34 for the isolates 2064 and 14733, respectively. Unlike *E. coli*, these *RMSE* values pointed out isolate variability. The lowest calculated *RMSE* values for populations in the background were also calculated for LAB in all combinations and ranged between 0.19 and 0.30 log CFU/mL. For *G. candidum*, the higher *RMSE* values of 0.51–0.63 log CFU/mL were calculated between their numbers in milk and milk with the addition of 1% NaCl, which also confirmed its sensitivity to NaCl.

Within the validation indices ($B_f$ and $A_f$), the fast-growing isolate 2064 model data and the experimental data of isolate 14733 were used. While the calculated $B_f$ values for the growth of *S. aureus* isolates were 1.184 and 1.284, the $A_f$ values were 1.371 and 1.500 in milk and 1% NaCl in milk, respectively.

LAB growth was accurately predicted in all cases of co-culture with *S. aureus* since the calculated $B_f$ values were within 0.980 to 1.016 and the $A_f$ values ranged from 1.023 to 1.035. Furthermore, the values of the interaction coefficient $I_{LS}$ varied between 0.91 and 1.26 (Table 3).

Referring to the interpretation of the $B_f$ when used for model performance evaluations involving pathogens, three categories were recommended by [55]. $B_f$ in the range of 0.90 to 1.05 can be considered good; 0.70 to 0.90 or 1.06 to 1.15 can be considered acceptable and less than 0.70 or greater than 1.15 should be considered unacceptable. In almost all cases with two isolates of each *E. coli* and *S. aureus* contaminant in our study, the values

of the $B_f$ factors were in the range of 0.90–1.05, which means that the H-GD model can be considered as suitable also for growth prediction in co-cultures with three different microbial populations.

### 4. Conclusions

The behaviour of microbial co-culture populations was successfully described with the H-GD model for growth in combination with a secondary Ratkowsky model. To describe the declination in the number of *E. coli* and *S. aureus*, the reparametrized Gompertz-inspired function with the Bigelow secondary model was used at temperatures related to the ripening of artisanal cheese. After the early stationary phase reached by LAB, *S. aureus* responded less competitively and more sensitively to the lactic acid produced by LAB than *E. coli*. However, the different behaviour of *E. coli* isolates found in this work may be associated more with different properties between isolates (competitiveness and sensitivity to NaCl) than with only variability, in general. Internal but cross-validation provided acceptable values for the predictions applicable in cheesemaking practise. LAB culture showed stable growth in all co-culture trials. *G. candidum* appears not to be inhibited by the presence of LAB, nor *E. coli* or *S. aureus,* and reached its typical maximum density in a stationary growth phase. The study can be used as a knowledge base for revisions of the hygiene guidelines for raw milk cheeses, as was our case in Slovakia. Furthermore, the fate of *E. coli* and *S. aureus* isolates within the interaction with other subpopulations during a short period of ripening can also be part of the exposure assessment. However, we expect that the complexity of the model, variability of the interactions themselves, and variability of the outputs are key factors and should be considered in future applications of this approach.

**Supplementary Materials:** The following supporting information can be downloaded at: https://www.mdpi.com/article/10.3390/app13158713/s1, Figure S1: Co-culture growth of LAB Fresco, *E. coli* PS2 and *G. candidum* in milk at 15, 18 and 21 °C (without and with 1% of NaCl, respectively); Figure S2: Co-culture growth of LAB Fresco, S. aureus 2064 and G. candidum in milk at 15 and 21 °C (without and with 1% NaCl, respectively).

**Author Contributions:** Conceptualization, Ľ.V.; Methodology, P.A., M.K. and Ľ.V.; Software, P.A.; Validation, P.A.; Investigation, M.K.; Writing—original draft, P.A.; Writing—review and editing, M.K. and Ľ.V.; Visualization, Ľ.V.; Supervision, Ľ.V.; Project administration, Ľ.V.; Funding acquisition, Ľ.V. All authors have read and agreed to the published version of the manuscript.

**Funding:** This research was funded by the Slovak Research and Development Agency grant number APVV 19-0031; Grant Agency of the Ministry of Education, Science and Research and Sport of the Slovak Republic grant number VEGA 1/0132/23.

**Institutional Review Board Statement:** Not applicable.

**Informed Consent Statement:** Not applicable.

**Data Availability Statement:** The data presented in this study are available on request from the corresponding author.

**Conflicts of Interest:** The authors declare no conflict of interest.

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
