# Peer review of "Modelling and Predicting the Growth of Escherichia coli and Staphylococcus aureus in Co-Culture with Geotrichum candidum and Lactic Acid Bacteria in Milk"

_applsci, doi:10.3390/app13158713_

Round 1
Reviewer 1 Report
The study reported an optimized model to research the competitive growth of co-culture species in milk.
Comments:
(1) Abstract. Key conclusion and quantitative results are lacked in this part. What’s the preferred temperature range for this model? What’s the advantage of this model (present with concrete data)? How can this model be applied in future food and milk industry? Potential readers will look for these in navigating papers.
(2) Introduction. Shorten this part to around 3-4 paragraphs, please focus more on the information of current models to assay and predict the growth variation of representative strains.
(3) Materials and methods. Please give one table to summarize all involved strains and their origins (model strains should have storage No.).
(4) Results and discussion. Expand the last paragraph of 3.1 as this is the key point of this study.
(5) Figures and tables. Improve the resolution and quality of all figures.
(6) Conclusion. Give some future prospects. Can this model be further applied in other cases? What’s the future trends in modelling methods in food microbiology?
Author Response
Response to Reviewer 1 Comments
Thank you very much for all your professional comments and suggestions on our manuscript applsci-2486690. After reading your suggestions, we were very grateful for your careful revision. We edited our manuscript based on your comments; all changes are highlighted in yellow within the revised manuscript.
Point 1: Abstract. Key conclusion and quantitative results are lacked in this part. What’s the preferred temperature range for this model? What’s the advantage of this model (present with concrete data)? How can this model be applied in future food and milk industry? Potential readers will look for these navigating papers.
Response 1: To be more specific, we added quantitative data referring to the behaviour of the most important population in cocultures. The application of the model was mentioned at the end of the abstract. However, in conclusions, we had to mention also the complexity of the model as a hurdle for application. The temperatures of incubation were also specified. We hope that the extended Abstract is now more suitable.
Point 2: Introduction. Shorten this part to around 3-4 paragraphs, please focus more on the information of current models to assay and predict the growth variation of representative strains.
Response 2: We revised and shortened the Introduction. One of the paragraphs was moved to the beginning of the Results and Discussion and a paragraph on the current model was added (actually it is the second paragraph if the introductory part).
Point 3: Materials and methods. Please give one table to summarize all involved strains and their origins (model strains should have storage No.).
Response 3: We summarized the used strains and mentioned their origin in the table. Please see Tab. 1. In association with this point, we replaced the term strain for isolate throughout the whole manuscript.
Point 4: Results and discussion. Expand the last paragraph of 3.1 as this is the key point of this study.
Response 4: We revised the Results and discussion in this context, please see the end of chapter 3.1
Point 5: Figures and tables. Improve the resolution and quality of all figures.
Response 5: All original figures in high resolution in Excel will be provided to the editor.
Point 6: Conclusion. Give some future prospects. Can this model be further applied in other cases? What’s the future trends in modelling methods in food microbiology?
Response 6: We revised the conclusions in this context, please see the yellow parts.
Reviewer 2 Report
The objective of the manuscript ‘Modelling and predicting the growth of Escherichia coli and Staphylococcus aureus in co-culture with Geotrichum candidum and lactic acid bacteria in milk’ was to investigate the growth of two pairs of co-cultures (Escherichia coli/Geotrichum candidum, and Staphylococcus aureus/Geotrichum candidum) with a starter culture of lactic acid bacteria . The topic undertaken in the study is of great interest in food safety and for public health. The authors designed a complex study and carried out an advanced mathematical analysis in predictive microbiology area. The theoretical background was described carefully and justifies the aim of work. Also methodology of the studies was described in detail. Additionally, I request the authors to provide information on the practical applicability of the developed models in industrial milk processing and cheese production, especially from raw milk.
A slight correction of English grammar is recommended.
Author Response
Response to Reviewer 2 Comments
Thank you very much for all your professional comments on our manuscript applsci-2486690. After reading it, we were very grateful for your revision.
Point 1:
Additionally, I request the authors to provide information on the practical applicability of the developed models in industrial milk processing and cheese production, especially from raw milk.
Response 1: Thank you for your suggestion. We hope that the extended conclusions and other parts on the yellow background bring lighter practical applicability to the model. However, we have to admit the complexity of the approach.
Reviewer 3 Report
Authors have previous publication in line with this. Please check below,
1. Ačai, P.; Valík, Ľ.; Medveďová, A. One- and Two-Step Kinetic Data Analysis Applied for Single and Co-Culture Growth of Staphylococcus aureus, Escherichia coli, and Lactic Acid Bacteria in Milk. Appl. Sci. 2021, 11, 8673. https://doi.org/10.3390/app11188673
2. Ačai P, Valík L, Medved'ová A, Rosskopf F. Modelling and predicting the simultaneous growth of Escherichia coli and lactic acid bacteria in milk. Food Sci Technol Int. 2016 Sep;22(6):475-84. doi: 10.1177/1082013215622840
3. Ačai P, Medved'ová A, Mančušková T, Valík L. Growth prediction of two bacterial populations in co-culture with lactic acid bacteria. Food Sci Technol Int. 2019 Dec;25(8):692-700. doi: 10.1177/1082013219860360.
At present, I am not able to see novelty component, not by methods, modelling and aim, thus I hereby recommend to reject the said manuscript.
Need improvements
Author Response
Response to Reviewer 3
Thank you for your valuable comments on our manuscript.
Point 1
At present, I am not able to see novelty component, not by methods,
modelling and aim, thus I hereby recommend to reject the said manuscript.
Response: We can understand your opinion. However, after already published works, we would like to mention that this is the last research we performed with co-cultures. In 2016, only E. coli and LAB were the subjects of modelling of microbial competition. In 2019, it was about two undesirable populations together with LAB, followed by a one-step approach that was applied in 2021. Moreover, the submitted article still provides some novelty points when compared to its predecessors. For example, new sets of data with the yeast G. candidum, one-step kinetic data analysis for the description of E. coli and S. aureus population declination (two isolates from each species within co-cultures in milk in the range of temperatures typical for cheese maturation). Additionally, for the first time, we have modelled and predicted the co-cultured growth of G. candidum under two different NaCl concentration conditions. However, we have to admit that the complexity of the approach can limit the practical applicability of the models for other users. Furthermore, we also added some arguments for taking our results into account by professionals, e.g., using the model in exposure assessments or data to revise hygiene guidelines for artisanal cheese production.
We hope that the changes performed in the manuscript will bring about a lighter explanation of our output.
Round 2
Reviewer 3 Report
Authors appropriately addressed the concerns raised by me, thus I hereby recommend to accept the said version of the manuscript.